# Study of Magnesium Formulations on Intestinal Cells to Influence Myometrium Cell Relaxation

**DOI:** 10.3390/nu12020573

**Published:** 2020-02-22

**Authors:** Francesca Uberti, Vera Morsanuto, Sara Ruga, Rebecca Galla, Mahitab Farghali, Felice Notte, Chiarella Bozzo, Corrado Magnani, Antonio Nardone, Claudio Molinari

**Affiliations:** 1Laboratory of Physiology, Department of Translational Medicine, University of Piemonte Orientale, via Solaroli 17, 28100 Novara, Italy; vera.morsanuto@uniupo.it (V.M.); sara.ruga@uniupo.it (S.R.); rebecca.galla@uniupo.it (R.G.); mahitab.farghali@uniupo.it (M.F.); felice.notte@uniupo.it (F.N.);; 2Laboratory of Applied Biology, Department of Translational Medicine, University of Piemonte Orientale, via Solaroli 17, 28100 Novara, Italy; chiarella.bozzo@uniupo.it; 3Medical Statistics and Cancer Epidemiology Unit, Department of Translational Medicine, University of Piemonte Orientale, 28100 Novara, Italy; corrado.magnani@uniupo.it; 4Department of Translational Medicine, University of Piemonte Orientale, 28100 Novara, Italy; antonio.nardone@uniupo.it

**Keywords:** Magnesium absorption, myometrial contractility, magnesium supplementation, PHM1- 41 cells, Caco-2 cells, magnesium mechanisms

## Abstract

*Background*: Magnesium is involved in a wide variety of physiological processes including direct relaxation of smooth muscle. A magnesium imbalance can be considered the primary cause or consequence of many pathophysiological conditions. The smooth muscle tissue of the uterus, i.e., the myometrium, undergoes numerous physiological changes during life, fundamental for uterine activities, and it receives proven benefits from magnesium supplementation. However, magnesium supplements have poor absorption and bioavailability. Furthermore, no data are available on the direct interaction between intestinal absorption of magnesium and relaxation of the myometrium. *Methods*: Permeability in human intestinal cells (Caco-2 cells) and direct effects on myometrial cells (PHM1-41 cells) of two different forms of magnesium, i.e., sucrosomial and bisglycinate, were studied in order to verify the magnesium capacity of modulate contractility. Cell viability, reactive oxygen species (ROS) and nitric oxide (NO) production, magnesium concentration, contractility, and pathways involved were analyzed. *Results*: Data showed a better influence of buffered chelate bisglycinate on intestinal permeability and myometrial relaxation over time with a maximum effect at 3 h and greater availability compared to the sucrosomial form. *Conclusions*: Magnesium-buffered bisglycinate chelate showed better intestinal absorption and myometrial contraction, indicating a better chance of effectiveness in human applications.

## 1. Introduction

Magnesium is the most important intracellular divalent cation; indeed, it is the most abundant cation in the cell after potassium. Magnesium plays a versatile biological role as it can serve structural functions, as well as dynamic functions [1]. For this reason, it is a key regulator of human health; it is involved in maintaining correct homeostasis in the body and it should be considered an aspecific ligand within the cell. Because of magnesium’s association with a wide range of metabolic processes, it is extremely important. However, an imbalance in the availability of magnesium can be considered the primary cause and/or consequence of pathophysiology conditions [2,3,4,5], including hypomagnesaemia [6]. To date, the potential efficacy of magnesium supplementation is widely recognized; however, oral magnesium supplementation exhibits poor absorption and bioavailability, due to the interference of dietary compounds or digestive factors [7]. Cereals, nuts, green vegetables, chocolate, legumes, and dairy products [8] contain magnesium, but most is lost during cooking or refining [9]. Magnesium homeostasis is closely checked by intestinal absorption of quantity contained in the food, exchange with bone, and renal excretion [10]. Magnesium enters the cell via channels or channel-like mechanisms. TRPM7 (Transient receptor potential cation channel, member 7) and TRPM6 (member 6) are identified as Mg^2+^ transport channels into mammalian cells, particularly at the intestinal and kidney levels [10]. These ion channels have a permeation profile of magnesium > calcium [11,12]. They are associated with Mg^2+^ influx and homeostasis [10,13]. While TRPM6 is mainly restricted to the absorptive epithelial intestine and kidney, TRPM7 is ubiquitously expressed and may be very important in regulating intracellular magnesium homeostasis and free intracellular levels of magnesium ([Mg^2+^]i) [11,14,15,16]. In addition, TRPM7 plays a double role within the cell by modulating Mg^2+^ homeostasis and cellular functions connected to cell adhesion, contractility, and the (anti)inflammatory process [17]. Among other transporters, MagT1 appears to possess high specificity for Mg^2+^ in humans; it is ubiquitously expressed [18,19] and involved in maintaining intracellular Mg^2+^ levels [20]. For this reason, enhanced bioavailability is likely to result in more effective supplementation of magnesium [7]. At the same time, magnesium has a direct smooth muscle relaxation effect [21], presumably acting as a calcium antagonist [22]. Myometrium is made up of smooth muscle tissue, which undergoes several physiological changes such as hypertrophy, hyperplasia, contraction, and apoptosis, which are critical for uterine activities [23].

Myometrium is influenced by several physiological mechanisms (neuronal, hormonal, metabolic, and mechanical) that regulate its activity during delivery. In obstetrics practice, drugs or phytochemicals are important for the alteration of uterine contractions, as they could lead to miscarriage or disruption of the normal course of parturition [24,25].

A main problem associated with uterine contractility concerns preterm delivery; the cause of death and morbidity of newly born babies is about 50% of all preterm deliveries due to preterm labor [26]. In spite of numerous published studies, the knowledge surrounding preterm study remains scarce because preterm labor has a multifactorial origin [27]. Furthermore, in in vitro fertilization (IVF) programs, preterm labor is a really common cause of serious problems [26]. In order to prevent preterm birth, the main processes that switch the myometrium from a quiescent state during pregnancy to an active state during pregnancy should be understood [28]. Since L-type Ca^2+^ channels are not involved in uterine smooth muscle cells, or they are partially inhibited by uterine contraction activators such as oxytocin, [Ca^2+^] is able to increase uterine contractions. [29]. cAMP content is increased by some agents (known as “cAMP agonists”) which can acutely inhibit myometrial contraction. According to this, PKA is suggested to induce relaxation of the myometrium not only via intracellular calcium mobilization inhibition, through phosphatidylinositol turnover inhibition and myosin light chain (MLC20) phosphorylation, but also via Rho signaling repression. cAMP’s ability to inhibit contractility could lead to considering myometrial cAMP signaling, via PKA activation, which is an effective approach to prevent preterm labor [30]. At a physiological level, uterine contractions are inhibited by extracellular Mg^2+^, which reportedly suppresses spontaneous depolarization. For this reason, Mg^2+^ is used in preterm labor to treat uterine contractions, but the mechanism of Mg^2+^ blocking uterine contractions is not known [29]. Recently, nitric oxide (NO), generated by nitric oxide synthase (NOS), was said to induce parturition in preterm labor since it has a relaxant effect on the myometrium. In addition, NOS activity is inhibited by magnesium concentration and, for that, it is extensively used for premature labor prevention [27]. Since no previous data are available on the direct interaction between magnesium intestinal absorption and myometrium relaxation, in the present study, we examine the permeability in human intestinal cells and the direct effect on myometrial cells, in order to verify magnesium’s contraction-inducing ability. Stimulated by the growing need for scientific data in this field, we decided to explore the effect of magnesium ion from magnesium bisglycinate chelate buffer (Trocà^®^ Mag) and sucrosomial magnesium (Ultramag^®^) on intestinal and myometrial cells.

## 2. Materials and Methods

### 2.1. Experimental Protocol

The Caco-2 cell line, supplied by the American Type Culture Collection (ATCC), was cultured in DMEM-F12 (Sigma-Aldrich, St. Louis, Stati Uniti) containing 10% FBS (Sigma-Aldrich), 2 mM l-glutamine (Sigma-Aldrich), and 1% penicillin–streptomycin (Sigma-Aldrich) at 37 °C in an incubator at 5% CO_2_ [31]. This cell line was used to understand the intestinal absorption following magnesium oral intake [32]. Caco-2 cells were plated at 1 × 10^4^ cells in 96-well plates to study cell viability via the MTT test, and by recording reactive oxygen species (ROS) and NO production; 5 × 10^4^ cells were plated on 24-well plates to analyze Ca^2+^ and Mg^2+^ currents; 2 × 10^4^ cells were plated on a 6.5-mm transwell with 0.4-μm pore polycarbonate membrane insert (Greiner bio-one, Kremsmünster, Austria) in a 24-well plate to investigate the absorption; 1 × 10^6^ cells were plated in six-well plates to study the intracellular mechanisms involved. The cells were used at passage numbers between 26 to 32 in order to maintain the physiological paracellular permeability and transport properties [33]. The PHM1-41 cell line, purchased from ATCC, was generated from myometrial cells obtained from term-pregnant myometrium; it was used as a “contraction model” [34] since this cell line maintains a primary phenotype [35]. PHM1-41 cells were maintained with high-glucose Dulbecco’s modified Eagle’s medium (DMEM, Sigma-Aldrich) conditioned with 0.1 mg/mL G-418, 2 mM glutamine, and 10% FBS. PHM1-41 cells were used between passages 20 and 25 to retain many morphological and phenotypic responses of primary myometrial cells [36];1 × 10^4^ cells were plated in 96-well plates to study cell viability via the MTT test, and by recording ROS and NO production; 5 × 10^4^ cells were plated on 24-well plates to analyze Ca^2+^ and Mg^2+^ currents; 1 × 10^6^ cells were plated in six-well plates to study intracellular mechanisms involved. 

In both cell types, cells were synchronized before stimulations, via overnight incubation (for Caco-2 cells) or incubation for 6 h (for PHM1-41 cells), with DMEM without red phenol (Sigma-Aldrich) and FBS, supplemented with 1% penicillin/streptomycin, 2 mM l-glutamine, and 1 mM sodium pyruvate in an incubator at 37 °C, 5% CO_2_, 95% humidity, and pre-treated for 2 h in HBSS buffer (1.26 mM CaCl_2_, 5.33 mM KCl, 0.441 mM KH_2_PO_4_, 4.17mM NaHCO_3_, 137.93 mM NaCl, 0.338 mM Na_2_HPO_4_, 5.56 mM d-glucose) to remove magnesium in the medium. Caco-2 cells were treated with 18% magnesium-buffered bisglycinate chelate (which contains 200 mg/g magnesium, Albion^®^, Balchem Corporation, New York, named MB, which is the main component of Trocà^®^ Mag, Laborest Italia srl, Italy) or sucrosomial magnesium (Ultramag^®^, Alesco, Pisa, named UM, which contains 333 mg/g magnesium) used at a final concentration of 1 mM, as reported in the literature [11]. Both magnesium forms were prepared in a 10× concentration directly in medium without adding other agents, according to the datasheet, which was directly used to obtain a final concentration of 1 mM in cells. PHM1-41 cells, seeded in a monolayer, were stimulated with pre-digested Caco-2 MB and UM, to verify the efficacy of both magnesium types after intestinal absorption.

### 2.2. Permeability Assay

Caco-2 cells, seeded on a transwell insert, were maintained in complete medium for 21 days [37,38] to induce differentiation of the cell, changing the culture medium every two days. After this time, cells were cultured under different pH conditions, with neutral pH (pH 7.4) in the basolateral chamber and acidic pH (pH 6.5) in the apical part [38,39]. Under this condition, both 1 mM MB and 1 mM UM were added to the apical environment in a time-course study (ranging from 1 h to 4 h), and, at each time point, the magnesium quantity at the basolateral level was determined by a Magnesium Assay Kit (Sigma-Aldrich), following the manufacturer’s instructions. 

### 2.3. Co-culture Model

Caco-2 cells were seeded into transwell inserts at a density of 20,000 cells/cm^2^ and were grown for 21 days in a complete medium, as previously reported. Then, 3–5 days before the deadline, PHM1-41 cells were added to the basolateral side of the insert system at a concentration of 50,000 cells/600 μL, and the co-culture was maintained in a complete medium for three days. Medium was changed starting from the apical side of the wells and incubated for up to 2 h without magnesium. After 2 h, the treatments were performed. Both 1 mM MB and 1 mM UM were added to the apical environment for 1 h and 3 h, and then total magnesium was measured by Magnesium Assay Kit at the basolateral level. At 4 h after stimulation, total magnesium was measured by the Magnesium Assay Kit in PHM1-41 lysates. PHM1-41 cells were lysed and homogenized in cold PBS 1× and centrifuged at 10,000 rpm for 10 min. After that, the supernatant was collected, and total Mg^2+^ concentration was measured.

### 2.4. Magnesium Assay Kit

The Magnesium Assay Kit (Sigma-Aldrich) was used to measure the amount of magnesium in culture supernatants (basolateral level of Caco-2 transwell) and lysate (in PHM1-41), following the manufacturer’s instructions [40]. Briefly, 50 μL of Master Mix Reaction Mix was added to the samples for 10 min at 37 °C, followed by covering the plates. The absorbance was measured by a spectrometer (VICTORX4, multilabel plate reader) at 450 nm. Magnesium concentrations can be determined using the standard curve, obtained from the kit instructions, and the concentration was expressed as Sa/Sv = C, in which Sa is the amount of magnesium in an unknown sample (nmol) from the standard curve, Sv is the sample volume (L) added to the reaction well, and C is the concentration of magnesium in our sample.

### 2.5. Cell Viability

An MTT-based in vitro Toxicology Assay Kit (Sigma-Aldrich) was applied to a 96 well-plate as previously described [39] (see Section A.1). 

### 2.6. ROS Production

Reactive oxygen species (ROS) were measured using a standard protocol [41] (see Section A.2).

### 2.7. NO Production

NO production was analyzed using the Griess Assay (Promega, Madison, Stati Uniti), as reported in the literature [42] (see Section A.3). 

### 2.8. Intracellular Magnesium

[Mg^2+^]i was determined using Mg^2+^-sensitive fluorescent dye Mag-fura-2-AM (Furaptra, Biotium). On the day of the experiment, both cell types were incubated in a Hanks salt solution (Thermo Fisher Scientific, Waltham, Stati Uniti) to maintain pH and osmotic balance without Mg^2+^, containing 10 mM glucose and supplemented with 20 mM HEPES/Tris (pH 7.4), 1.3 mM CaCl_2_, and 5 μM Mag-fura-2-AM at 37 °C for 30 min. The fluorescence of loaded cells was monitored for 3 h at regular intervals (10 min, 30 min, 1 h, and 3 h). Fluorescence of Mag-fura-2 with excitation at 340 nm and 380 nm was acquired at the wavelength of emission of 510 nm with an exposure time of 100 ms, using a Fluorescence Spectrometer VICTOR X4. After subtracting the background, the fluorescence ratios (340/380 nm) were calculated and compared to control. The Rmax for Mag-fura-2 was analyzed by the addition of 50 mM MgCl_2_, and Rmin was determined by removal of Mg^2+^ and the addition of 100 mM EDTA [10].

### 2.9. [Ca^2+^] Measurement

Both cell types were washed twice with sterile PBS 1× and incubated with 5 μM Fura-2 AM (Sigma-Aldrich) for 30 min in the dark in PSS buffer without Mg^2+^ (1.5 mM KCl, 10 mM HEPES, 10 mM d-Glucose, 2 mM l-Glutamine, pH 7.4), in agitation at 37 °C. Magnesium (UM and MB) was added to the suspension of Fura-2/AM loaded cells, and the fluorescence was measured by the fluorescence spectrometer (VICTOR X4) at the wavelengths of 340 nm for excitation and 510 nm for emission. The results were expressed as a percentage compared to control cells. 

### 2.10. Western Blot of Cell Lysates

Caco-2 and PHM1-41 cells were lysed in ice with Complete Tablet Buffer (Roche, Basilea, Svizzera) supplemented with 2 mM sodium orthovanadate (Na_3_VO_4_), 1 mM phenylmethanesulfonyl fluoride (PMSF) (Sigma-Aldrich), 1:50 mix Phosphatase Inhibitor Cocktail (Sigma-Aldrich), and 1:200 mix Protease Inhibitor Cocktail (Sigma-Aldrich). According to the standard protocol, 35 μg of protein of each sample was resolved on 8% or 15% SDS-PAGE gels, and polyvinylidene difluoride membranes (PVDF, GE, Healthcare Europe GmbH) were incubated overnight at 4 °C with the following specific primary antibodies: anti-Phospho-PKAα/β/γ (1:250; Santa Cruz, California, Stati Uniti), anti-Phospho-p44/p42 mitogen-activated protein kinase (1:1000; Cell Signaling Technology), anti-ERK/MAPK (1:1000; Cell Signaling Technology, Danvers, Stati Uniti), anti-Phospho-Ca^2+^/calmodulin-dependent protein kinase IIα (1:250; Santa Cruz), anti-Occludin (1:500; Arigo Biolaboratories, Hsinchu, Taiwan), anti-Claudin 4 (1:1000; Santa Cruz), anti-Zona occludens protein 1 (1:1000; DBA, Segrate, Italia), anti-TRMP7 (1:1000; Santa Cruz), anti-magnesium transporter 1 (1:1000; DBA Italia). In addition, only for PHM1-41 cells, the following antibodies were used: anti-phospho-Akt (1:1000; Cell Signaling Technology), anti-Akt (1:500; Cell Signaling Technology), anti-NOS2 (1:250; Santa Cruz), and anti-Desmin (1:1000; Santa Cruz). Protein expression was normalized and verified through anti-β-actin (Sigma-Aldrich). The results were expressed as means ± SD (% vs. control). 

### 2.11. Animal Model 

Uterine myometrial samples for isometric contraction measurements were obtained from adult (8–12 weeks old) CD1 wild-type mice on day 19 of pregnancy. These experiments were performed on nine pregnant mice. Mice were purchased from Charles River Laboratories (Calco, Italy), the animals had ad libitum access to food and water, and they were housed in a 12-h light/12-h dark cycle. All experimental procedures on animals were approved by the Internal University Committee OPBA (Organismo preposto al benessere degli animali) in accordance with local ethical standards and protocols. Experimental protocols were approved by national guidelines (Ministero della Salute authorization number 914/2015-PR) and in accordance with the Guide for the Care and Use of Laboratory Animals (National Institutes of Health publication 86–23, 1985 revision). The animals were euthanized by cervical dislocation under a lethal dose of isoflurane, and uterine myometrial samples were removed to be processed for the uterine isometric contractility assay. 

### 2.12. Uterine Isometric Contractility Assay 

Isometric contraction measurement was performed as reported in the literature [43]. Briefly, uterine myometrial samples, obtained from wild-type mice on day 19 of pregnancy, were prepared by cutting the uterus along the mesometrial (vascularized) border, followed by removal of fetal, placental, and connective tissue/membranes obtaining uterine strips of 4 mm × 12 mm. A total of 36 uterine strips was obtained from nine mice [44]. The sample size was assessed adequately using a preliminary statistical analysis (G*Power software). Each strip was placed under 1 *g* tension for 60 min prior to recording baseline spontaneous contractile activity in Krebs solution. To verify the contraction, caffeine (1 mM) was added to organ baths and 10^−6^ M oxytocin was used to verify the maximal uterine contraction. To determine the effects of Mg at 1 and 3 h, the uterine strip in each organ bath chamber was incubated after stimulation with caffeine for 1 h. Oxytocin AUC was measured at 20-min intervals from oxytocin application and normalized to the tissue weight in grams (g). All treatment data were then expressed as a percentage compared to the maximal uterine contraction obtained using oxytocin. Data were analyzed using Prism software [43]. 

### 2.13. Statistical Analysis

Data were processed using Prism GraphPad statistical software, and one-way analysis of variance (ANOVA), followed by Bonferroni post hoc tests, was carried out for statistical analysis to compare groups. All results were expressed as means ± SD of four independent experiments performed in four technical replicates; in the stretch test, data were expressed as means ± SD of three independent experiments. Differences were considered to be statistically significant with a *p*-value < 0.05.

## 3. Results

### 3.1. Dose–response and Time-dependent Study of Cell Viability on Caco-2 Cells

Since magnesium formulations for human use are numerous and could affect cell viability, the MTT cell viability assay was performed. The first set of experiments was performed to demonstrate the effect on cell viability exerted by direct exposure to different concentrations of UM and MB over time. As shown in Table A1 (Appendix B), both UM and MB were able to induce a dose-dependent increase in cell viability starting from 1 h until 6h (*p* < 0.05 vs. control) with a greater effect at 1 mM compared to other concentrations (2.5 mM and 5mM). Furthermore, 1mM MB appeared to have a significantly greater effect (*p* < 0.05 vs. UM) than UM during the analyzed period (ranging from 1 h to 6 h) with a peak of viability at 3 h. All these data confirmed that neither magnesium form had a cytotoxic effect nor a time-dependent effect on Caco-2 cells.

### 3.2. Time-dependent Permeability after Stimulations of Caco-2 Cells with UM and MB

In order to study the biological functions of UM and MB, some experiments were performed on Caco-2 in a transwell transporting set-up to evaluate the Mg^2+^ intestinal absorption. The analysis of the basolateral environment (Mg^2+^ crossing the intestinal membrane to enter blood circulation) showed that both UM and MB had a time-dependent absorption starting from 1 h to 4 h compared to control (*p* < 0.05), as reported in Figure 1A. In addition, the amount of MB was higher than UM along the analyzed period (*p* < 0.05), with a greater effect at 3 h, in which the concentration of Mg^2+^ formulated in MB was 64% compared to UM (*p* < 0.05). These data support the hypothesis that the permeability of MB was higher than that of UM during the intestinal emptying time (ranging from 1 h to 4 h). However, only the apical to basolateral transport was evaluated, which could not indicate the mechanism of absorption involved. Moreover, the cells used exhibited tight junctions, indicating a rapid permeation. Since the main absorption time for both Mg forms was observed at 3 h, at this time point, ROS and NO productions were also investigated at the apical level (Figure 1B,C) in order to exclude any intestinal radical imbalance. Under physiological conditions, these two parameters should be balanced; ROS and NO levels produced by MB were lower than those produced by UM (*p* < 0.05, five-fold and 4.5-fold lower, respectively), indicating no inside effects during treatment with MB. These data support previous findings regarding the better cell viability and absorption of MB compared to UM.

### 3.3. Analysis of Permeabilization Mechanism on Caco-2 Cells Treated with UM and MB

Since Mg^2+^ homeostasis is tightly controlled by the dynamic action of intestinal absorption, TRMP7, a channel kinase involved in the active transcellular Mg transport processes in intestine, and MagT1, a selective Mg^2+^ transport protein also involved in maintaining intracellular Mg^2+^ levels, were also investigated on Caco-2 cells treated with UM and MB for 3 h (time with the highest absorption rate). As reported in Figure 1D,E, MB was able to induce the expressions, measured by Western blot and densitometric analysis, of both Mg^2+^ transporters with a greater effect compared to UM (*p* < 0.05, 80% and 650%, respectively) and to control (*p* < 0.05), supporting the extrusion mechanism hypothesized during the permeability assay. These data are important to demonstrate a better bioavailability of MB compared to UM. In addition, in order to confirm the tight junction activity of cells (physiological role during the intracellular passage of Mg^2+^), the effects of both UM and MB were analyzed by Western blot and densitometric analysis on occludin, claudin 4, and Zo-1 expressions. As demonstrated by data reported in Figure 2A–C, MB was able to induce a greater expression of all analyzed parameters (*p* < 0.05) compared to control and to UM (40%, 12%, and 47%, respectively). These findings demonstrate, for the first time, the influence of both Mg forms on permeability markers. Indeed, the effect of MB on occludin (Figure 2A) indicates a better regulation of barrier permeability and stimulation to cross the barrier by Mg2+; the effect of MB on claudin 4 (Figure 2B) indicates correct epithelial permeability mediated by a correct mechanism, regulated by tight junctions at the intracellular level; the effect of MB on Zo-1 (Figure 2C) demonstrates a better influence of this Mg formulation going from the cytoplasmic level to the transduction signal at the cell-to-cell junction. All these findings support the results observed before and indicate some effects on Caco-2 cells at the intracellular level, such as the mechanism at the basis of intestinal peristalsis, in addition to permeability.

### 3.4. Magnesium Flux and Its Relative Intracellular Pathways Activated in Caco-2 Cells

In order to clarify the importance of Mg^2+^ at the intestinal level, additional experiments were carried out to analyze the Mg^2+^ and Ca2+ movements after treatments with UM and MB under free- Mg^2+^ medium conditions. [Mg^2+^]i determined by Furaptra analysis (Figure 3A) showed a time- dependent influx of Mg2+ starting from 10 min to 3 h (the time of greatest absorption observed before) in the presence of both UM and MB compared to control (*p* < 0.05). In addition, both Mg forms had a physiological curve (up and down which were supposedly related to peristaltic movement). However, between UM and MB, the latter seemed to have greater effects over time which were significant (*p* < 0.05) only at 10 min and 3 h (67% and 133%, respectively), supporting the better influence of MB compared to UM as observed before. In addition, these effects of both Mg forms appeared to regulate a physiological balance between [Mg^2+^]i and Ca^2+^ flux; indeed, [Ca^2+^] (Figure 3B) had an opposite curve after treatments with UM and MB. In particular, a greater difference between MB and UM was observed at 10 min and 3 h (*p* < 0.05). These data confirm the beneficial role exerted by UM and MB to support intestinal motility, although MB appears to have a greater influence than UM. 

Since the Mg^2+^ and Ca^2+^ equilibrium is favored by some kinases, further experiments were carried out to study the involvement of PKA, which is present in the production of cAMP caused by Mg^2+^, ERK/MAPKs, which are important for cell viability and in calcium-dependent mechanisms such as the contraction-relaxation cycle, and CamKII, which is involved in calcium flow. Western blot and densitometric analysis (Figure 4A–C) were performed. In all the analyzed parameters, MB showed greater effects than UM (about 1.7-fold in A, about eight-fold in B, and about two-fold in C), supporting the previous data observed on the cells. These results are important in exploring the specific mechanism activated by UM and MB and the possible human applications to maximize the beneficial effects of Mg^2+^.

### 3.5. Dose–response and Time-dependent Studies on Cell Viability in PHM1-41 Cells

A new important application of Mg^2+^ involves inducing myometrial relaxation in order to prevent preterm labor. For this reason, UM and MB were tested on human myometrial cells (PHM1-41 cells) to verify their possible applications after intestinal digestion of both magnesium forms at 1 mM. The choice of this concentration was based on a dose–response study on intestinal cells, which mimics human oral intake. Cell viability after UM and MB (Table A2, Appendix B) administration showed a time-dependent effect with a peak at 3 h for both Mg forms. In addition, both UM and MB induced a significant effect on cell viability (*p* < 0.05) starting from 1 h, compared to control, indicating a positive influence of Mg^2+^ on myometrial cells. However, MB appeared to have a greatly significant effect on cell viability during all analyzed periods compared to UM (*p* < 0.05), which was also reported in Caco-2 experiments.

### 3.6. Time-dependent Effects of UM and MB in PHM1-41

In order to explore the influence of Mg forms on myometrial cells after intestinal absorption, in the first set of experiments, a co-culture model using Caco-2 and PHM1-41 cells was used to quantify the presence of Mg^2+^ crossed during time. Subsequently, PHM1-41 cells only were used to measure the intracellular Mg^2+^ rate. As reported in Figure 5A, both UM and MB were able to act after intestinal passage in a time-dependent manner with main effects at 3 h (about 27% and 44%, respectively) compared to control (*p* < 0.05). In detail, MB seemed to have its main effect during all periods analyzed (*p* < 0.05) compared to UM. In addition, as shown in Figure 5B, the intracellular level of Mg^2+^ measured in PHM1-41 cells over time (ranging from 1 h to 4 h) supported the results obtained in the co-culture model, indicating a time-dependent effect with a main peak at 3 h for both Mg forms. Finally, MB had significant effects during all analyzed periods (*p* < 0.05) compared to UM, indicating a better compliance over time. Main effects were observed in both UM and MB treatments at 3 h; thus, all successive experiments were performed at this very time point. Indeed, in the second set of experiments, the analyses of ROS and NO in PHM1-41 cells were carried out at 3 h of stimulation. As reported in Figure 5C, ROS produced by UM at 3 h was higher than that one produced by MB (*p* < 0.05), supporting less viability observed with UM compared to MB. On the other hand, NO produced by MB was higher than UM at 3 h (about three-fold, *p* < 0.05; Figure 5D), supporting data present in the literature on the importance of NO during myometrial relaxation. All these findings support the hypothesis of a possible use of UM and MB to improve human myometrial relaxation.

### 3.7. Relaxing Effects of UM and MB on PHM1-41

Additional experiments were carried out to analyze Mg^2+^ and Ca^2+^ movements after treatments with UM and MB under Mg^2+^-free medium conditions. [Mg^2+^]i determined by Furaptra analysis (Figure 6A) showed a time-dependent influx of Mg^2+^ starting from 10 min to 3 h, i.e., the time of greatest effect observed before in the presence of both UM and MB compared to control (*p* < 0.05). In addition, MB seemed to have greater effects over time starting from 1 h (*p* < 0.05) compared to UM, supporting the quantification analysis observed before. In addition, these effects of both Mg forms appeared to have a physiological balance with Ca^2+^ flux; indeed, [Ca^2+^] (Figure 6B) had an opposite curve after treatments with UM and MB. These data show the beneficial role exerted by UM and MB in supporting myometrial relaxation, although MB appeared to have a greater influence than UM. These results were also confirmed by the preliminary data obtained from the analysis of contractility on mouse strips using a uterine isometric contractility assay (Figure 6C). 

In order to clarify the mechanisms underlying the results observed, a selective Mg^2+^ transport protein involved in maintaining intracellular Mg^2+^ levels was investigated by Western blot and densitometric analysis (Figure 7A). The MagT1 expression induced by MB was higher than that induced by UM (*p* < 0.05), supporting better results regarding the absorption rate of Mg^2+^. In addition, the expressions of iNOS (Figure 7B) confirmed the role of NO in myometrial relaxation induced by UM and MB compared to control (*p* < 0.05), but MB appeared to have a greater effect on this kinase compared to UM (about 67%, *p* < 0.05), confirming its greater influence. Finally, the balance between ERK/MAPK and PI3K/Akt activities was also investigated to exclude the excitatory action of Mg forms. As illustrated in Figure 7C,D, their expressions reflected the correct balance to induce relaxation; when ERKs were up, Akt was down, indicating only an influence on viability. Indeed, both UM and MB were able to maintain the balance of expression needed to induce relaxation, but MB had a greater effect compared to UM on both kinases (*p* < 0.05).

Finally, PKA and phospho-myosin light chain (p-MLCK) were also investigated by Western blot and densitometric analysis (Figure 8). The expression of PKA (Figure 8A) is important to investigate the mechanism of relaxation; indeed, its expression induces cAMP production, which is able to improve the myometrial relaxation. UM and MB were able to induce the expression of PKA at 3 h of stimulation, and MB had a greater effect compared to UM (*p* < 0.05), supporting its major role in relaxation. On the contrary, p-MLCK (Figure 8B) may be downregulated to confirm the relaxation; however, both UM and MB maintained it at the basal level, and MB showed a significant effect. Our data explain the relaxation mechanism induced by both UM and MB, confirming the better ability of MB to act as a relaxing agent.

## 4. Discussion

This work aimed to evaluate the absorption mechanisms of Mg^2+^ administered in two different formulations. At the same time, knowledge concerning the passage of Mg^2+^ through enterocytes into the bloodstream was improved. One of the concepts not yet fully understood is the use of metabolic energy by cells to control Mg^2+^ uptake. While no metabolic energy is needed for Mg uptake as the Mg concentration in the intestinal cells is lower than that in the intestinal tract after meal intake, cellular extrusion of Mg is energy-dependent. In the literature, some authors [45] suggested a combination of a saturable (transcellular) mechanism and passive (paracellular) diffusion dominating at higher Mg concentrations to explain the mechanism. The existence of an active transport mechanism across the human epithelium of the intestine, using specific active carriers [46], was suggested. The nature of the saturable process is not well understood. A saturable mechanism could be based on active (carrier-mediated) transcellular transport or facilitated diffusion, e.g., mediated by passive carrier proteins, as the permeability of the membranes is, otherwise, low [47]. Several transport mechanisms were proposed, e.g., a proton-driven luminal Mg^2+^ carrier or channel [48]. A big concentration of magnesium is absorbed in the small intestine via a passive paracellular mechanism, which relies on tight junction permeability [39]. Magnesium regulatory fraction is transported via transcellular transporter TRMP7, a member of the long transient receptor potential channel family, which also plays an important role in intestinal calcium absorption [49]. Starting from these studies, the mechanisms involving TRPM7, MagT1, and tight junctions (in particular occludin, claudin 4, and Zo-1) were also investigated under both UM and MB stimulations. As demonstrated by the experiments described above, both Mg forms were able to induce all necessary protein expressions to support the mechanism of intestinal absorption. In addition, between UM and MB, the latter appeared to have greater effects compared to UM, indicating a better absorption at the intestinal level. In this context, we can hypothesize a possibly greater availability of MB compared to UM, which was confirmed by successive experiments on magnesium and calcium fluxes. Signaling, driven by PKA, is known to regulate the assembly and the opening of the paracellular route in epithelial cells. Moreover, MAP signaling pathways could modulate tight junction paracellular transport via up- or downregulation of several tight junction protein expressions [50]. In these experiments with both UM and MB, the expressions of these kinases were also confirmed, indicating a shared mechanism used by both Mg forms to cross the intestinal barrier. However, MB had greater effects compared to UM on the kinase expressions, confirming its better effectiveness. 

Nevertheless, at the present time, there is no agreement on which single evaluation system should be used to study Mg absorption in human subjects. Several studies are currently focusing on magnesium supplementation in different diseases [51]. However, conventional oral magnesium supplementation exhibits poor absorption and bioavailability. For this reason, enhanced bioavailability would likely result in more effective supplementation of magnesium [52]. Indeed, UM and MB have more efficiency to cross the intestinal barrier compared to magnesium citrate or oxide as described by blood concentration reported in another study, in which MB confirmed its effects over time [52]. 

Magnesium is also well known to have a relaxant effect on uterine muscle and, as such, it is used as a tocolytic agent in preterm labor. Magnesium is also used to prevent leg cramps during pregnancy [53] and as a tocolytic agent in preterm labor due to its relaxant effect on uterine muscle [54]. It seems that, in vitro, the magnesium ion is only strongly efficient at therapeutic concentrations (below 2.5 mM) on the spontaneous pregnant myometrial contractile activity. This effect follows a linear correlation with magnesium concentrations [22]. Magnesium tocolytic properties are similar to what was studied, with similar results in some human clinical studies on magnesium sulfate [55]. However, no one studied the molecular mechanisms of muscle cells to clarify whether Mg supplement is able to reach the target tissue and through which mechanism it exerts beneficial effects. In this context, the time and dose of UM and MB were also assessed on myometrial human cells, PHM1-41, evaluating cell viability. UM and MB were confirmed to have a time-dependent effect, with their main effect at 3 h of stimulation after intestinal digestion, but MB had a greater effect compared to UM, confirming its better compliance.

Extracellular Mg is able to inhibit uterine contractions, and it reportedly suppresses spontaneous depolarization at physiological concentrations. However, although Mg^2+^ is used clinically to treat uterine contractions in preterm labor, the mechanism of Mg blocking uterine contractions is not well understood [29]. The present study demonstrates that, in PHM1-41 cells, UM and MB are able to enter the intracellular environment over time with a maximum effect at 3 h. In addition, MB is able to reach a higher concentration compared to UM, supporting the hypothesis of better availability. The relaxing capacity of magnesium was also supported by preliminary results obtained from the analysis of contractility on mouse uterine strips using an isometric contractility assay.

Another important element in maintaining the myometrial relaxation is a correct balance between ROS and NO productions. Indeed, when ROS production exceeds the myometrial antioxidant scavenging capacity, oxidative stress may increase due to consequent cellular dysfunction and tissue damage [56]. On the other hand, NO is clearly an effective agent in relaxing spontaneous contractions in both animal and human myometrium [57]. Recently, a great deal of interest was also devoted to the relaxant effect of NO on the myometrium. Therefore, NO relaxes the myometrium via an increase in Mg and an inhibition of Ca^2+^ [27]. As illustrated by results obtained from PHM1-41 cells, UM and MB were able to maintain a balance between NO and ROS production, but MB seemed to be a better choice because it was able to induce a minor ROS and a major NO production compared to UM, indicating a better relaxing property. 

This fact was also confirmed by iNOS expression, which is a kinase that induces NO production; MB is able to induce a higher expression compared to UM. However, the importance of the mechanism underlying myometrial relaxation should be further investigated. Recently, studies on the mechanisms of action of magnesium during the inhibition of myometrial contractions reported that it rapidly induces a dose-related inhibition of the uptake of Ca^2+^. Researchers concluded that this inhibition of calcium uptake is responsible for the inhibitory effects of magnesium on spontaneous myometrial contractions [58]. The analysis of a time-course study performed on PHM1-41 cells to analyze Ca^2+^ and Mg^2+^ fluxes confirmed this mechanism, indicating that both UM and MB are able to induce myometrial relaxation over time with a greater effect at 3 h. Between the two Mg forms, MB confirmed its better influence compared to UM. This finding was also confirmed by the expression of MagT1, which was induced by both UM and MB, but the latter seems to have a greater effect compared to UM, supporting its better absorption. In addition, ERK/MAPKs, following the intracellular Ca^2+^ concentration, can induce contraction or relaxation on myometrial tissue [59].

Indeed, on PHM1-41 cells, the expression of ERKs is higher for MB than UM, but in balance with PI3K/Akt expression which was maintained at a basal level. This indicates the presence of a relaxing mechanism exerted by MB on myometrial cells. The key components that promote myometrial cAMP signaling are reduced upon labor onset, such as PKA, which promotes smooth muscle relaxation not only in the myometrium, but also in the myocardium and airways smooth muscle [30]. Furthermore, in this context, MB was able to induce the expression of PKA better than UM in PHM1-41 cells, indicating a major production of cAMP, which is important to induce relaxation. Finally, in order to exclude any contractile activity, the myosin light chain kinase (MLCK) marker [60] was also investigated. As illustrated by Western blot experiments in PHM1-41 cells, neither UM nor MB induced the phosphorylation of this marker, but MB had a major relaxing effect, supported by a major de-phosphorylation of MLCK.

Regarding the significance of this study regarding the in vivo applicability of the results, it is to be noted that the Caco-2 cell model allows calculating the permeability values that correlate well with human in vivo absorption data for different drugs and chemicals following the modern concept of translational medicine. As a result, the use of the Caco-2 model to predict human oral absorption of substances and ingredients is growing, in addition to its importance as a quality screening tool for intestinal absorption, especially in drug research.

## 5. Conclusions

In summary, the effect of Trocà^®^ Mag and Ultramag^®^ on both Caco-2 and PHM1-41 cells was analyzed to study how the different formulations of magnesium can influence its intestinal absorption and myometrial contraction. Although statistically significant effects were found for both formulations, the results showed a better influence of Trocà^®^ Mag on intestinal absorption, indicating a better chance of effectiveness in human applications.

## Figures and Tables

**Figure 1 nutrients-12-00573-f001:**
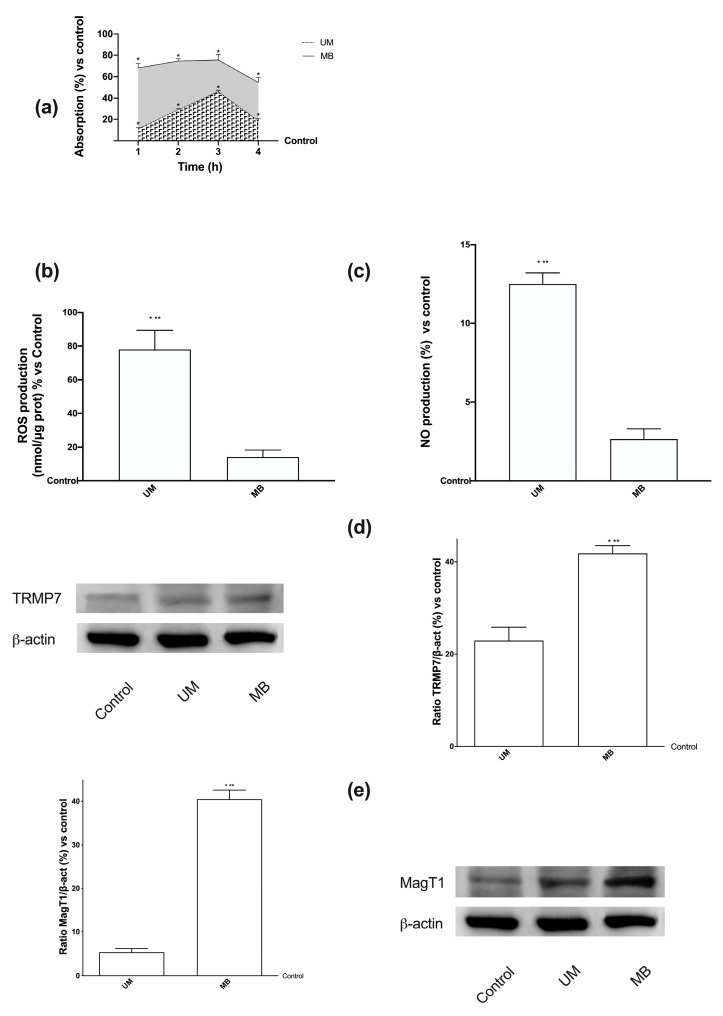
Magnesium and magnesium transport quantification, and balance of reactive oxygen species (ROS)/nitric oxide (NO) produced on Caco-2 cells. (**a**) Total Mg absorbed measured at the basolateral level on transwell during time (ranging from 1 h to 4 h). Data are means ± SD (%) compared to control values (line 0%) of four independent experiments produced in triplicate. * *p* < 0.05 vs. control; *p* < 0.05 between sucrosomial magnesium (UM) and magnesium-buffered bisglycinate chelate (MB) at the same time point across all time points. (**b**) ROS analysis measured at 3 h expressed as means ± SD (%) of cytochrome C reduced/µg of protein normalized to control (line 0%) of five independent experiments produced in triplicate. * *p* < 0.05 vs. control; ** *p* < 0.05 vs. MB. (**c**) NO production measured at 3 h normalized to control (line 0%) and expressed as means ± SD (%) of five independent experiments produced in triplicate. * *p* < 0.05 vs. control; ** *p* < 0.05 vs. MB. The images reported in (**d**) and (**e**) are examples of each protein of five independent experiments reproduced in triplicate. (**d,e**) Densitometric analysis of TRPM7 and MagT1 expression obtained in whole Caco-2 lysates at 3 h of stimulation. Data are expressed as means ± SD (%) of five independent experiments normalized and verified on β-actin detection. * *p* < 0.05 vs. control; ** *p* < 0.05 vs. UM.

**Figure 2 nutrients-12-00573-f002:**
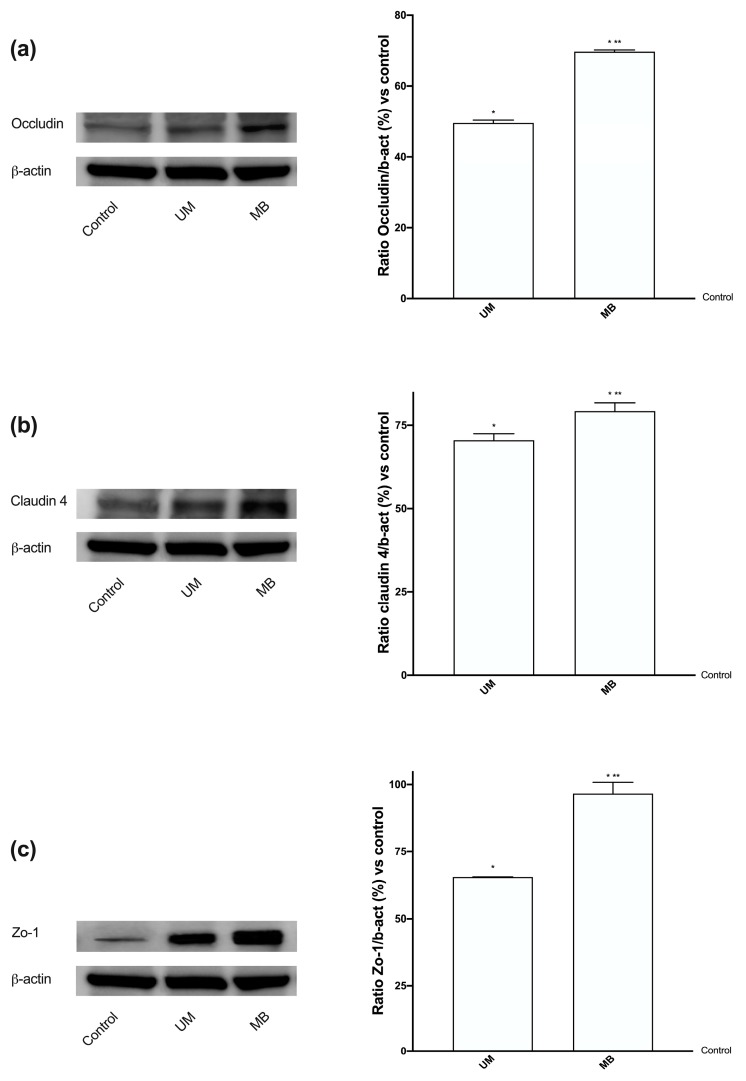
Western blot and densitometric analysis of some tight junctions on Caco-2 cells. Western blot and densitometric analysis of occludin (**a**), claudin 4 (**b**), and Zo-1 (**c**), analyzed on whole Caco-2 lysates at 3 h of stimulations. The images shown are an example of each protein from five independent experiments reproduced in triplicate. Data are expressed as means ± SD (%) of five independent experiments normalized and verified on β-actin detection. The abbreviations are the same as reported in Figure 1. * *p* < 0.05 vs. control; ** *p* < 0.05 vs. UM.

**Figure 3 nutrients-12-00573-f003:**
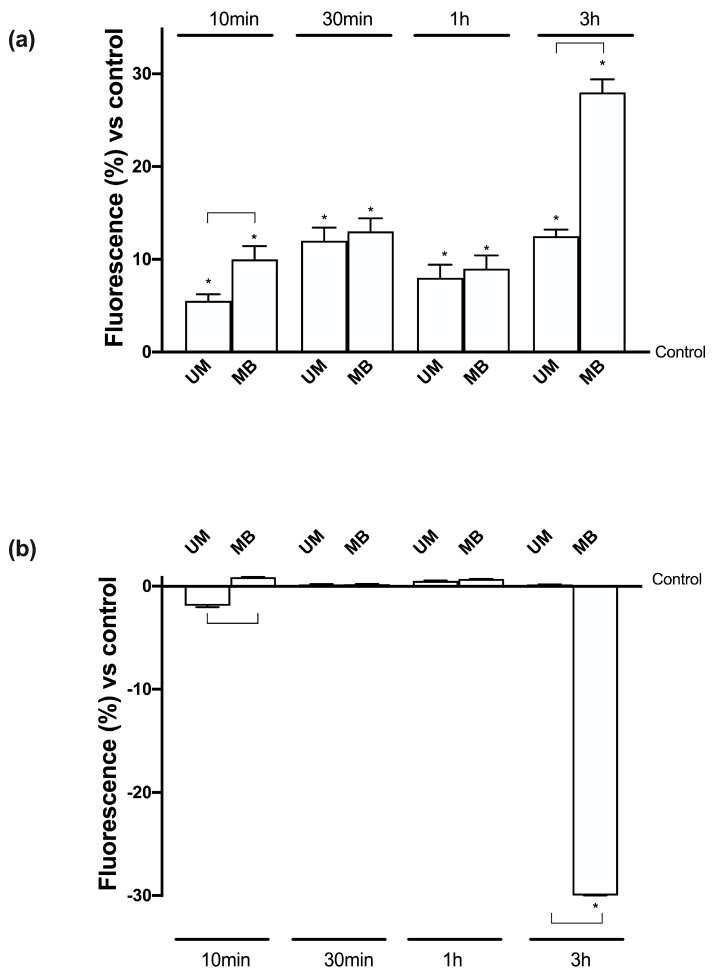
Measurements of Mg^2+^ and Ca^2+^ flux on Caco-2 cells. (**a**) Free intracellular magnesium concentration ([Mg^2+^]i) as measured before (basal [Mg^2+^]i) and after Mg-loading by UM and MB. After measurement of the basal [Mg^2+^]i in Na- and Mg-free medium, Caco-2 cells were Mg-loaded by Furaptra, and post-loading [Mg^2+^]i was calculated from the values measured over time. (**b**) Free calcium concentration ([Ca^2+^]) as measured before (basal) and after Mg-loading by UM and MB. After measurement of the basal [Ca^2+^] in Na-, Ca-, and Mg-free medium, Caco-2 cells were loaded with by Fura2, and post-loading [Ca^2+^] was calculated from the values measured during time. Values are means ± SD (%) of six single experiments. * *p* < 0.05 vs. control; the bars indicate *p* < 0.05 between UM and MB at the same time point.

**Figure 4 nutrients-12-00573-f004:**
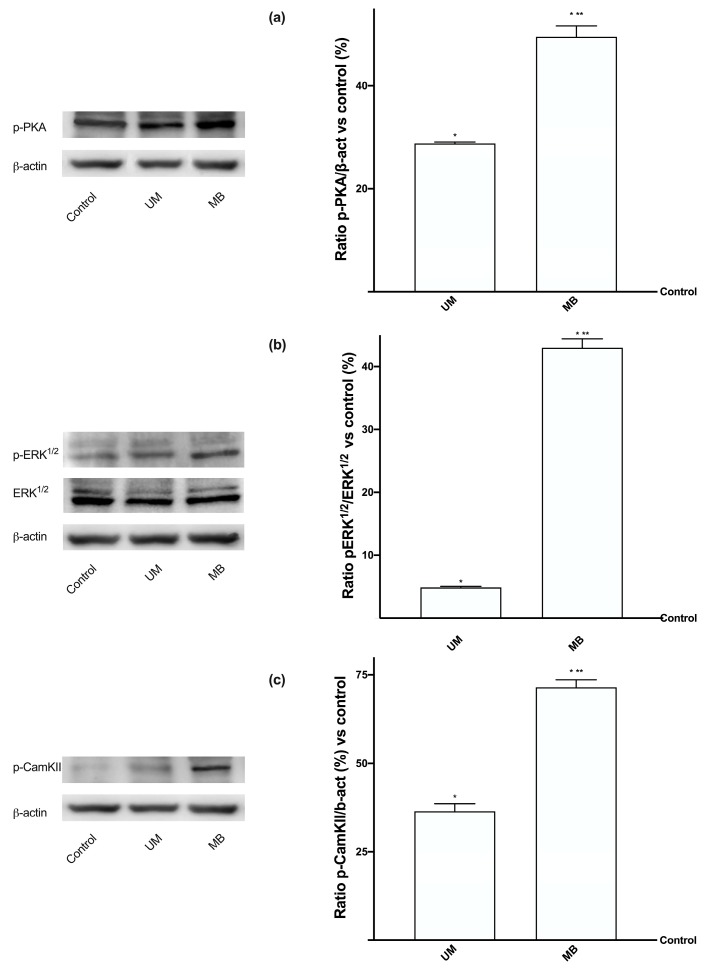
Western blot and densitometric analysis of PKA, ERK/MAPK, and CamKII on Caco-2 cells. Western blot and densitometric analysis of PKA (**a**), ERK/MAPK (**b**), and CamKII (**c**), analyzed on whole Caco-2 lysates at 3 h of stimulations. The images reported are an example of each protein from five independent experiments reproduced in triplicate. Data are expressed as means ± SD (%) of five independent experiments normalized on specific total protein when possible and verified on β-actin detection. The abbreviations are the same as reported in Figure 1. * *p* < 0.05 vs. control; ** *p* < 0.05 vs. UM.

**Figure 5 nutrients-12-00573-f005:**
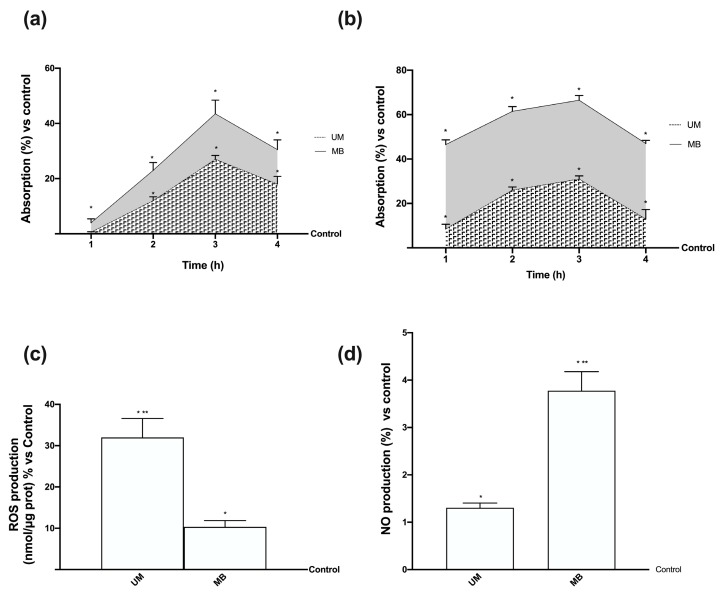
Magnesium quantification and ROS/NO balance produced on PHM1-41 cells. (**a**) Total Mg absorbed in co-culture model measured at the basolateral level over time (ranging from 1 h to 4 h). (**b**) Total Mg absorbed at the intracellular level over time (ranging from 1 h to 4 h). Data are means ± SD (%) compared to control values (line 0%) of four independent experiments produced in triplicate. * *p* < 0.05 vs. control; *p* < 0.05 between UM and MB at the same time point across all time points. (**c**) ROS analysis measured at 3 h expressed as means ± SD (%) of cytochrome C reduced/µg of protein normalized to control (line 0%) of five independent experiments produced in triplicate. * *p* < 0.05 vs. control; ** *p* < 0.05 vs. MB. (**d**) NO production measured at 3 h normalized to control (line 0%) and expressed as means ± SD (%) of five independent experiments produced in triplicate. * *p* < 0.05 vs. control; ** *p* < 0.05 vs. MB. The abbreviations are the same as reported in Figure 1.

**Figure 6 nutrients-12-00573-f006:**
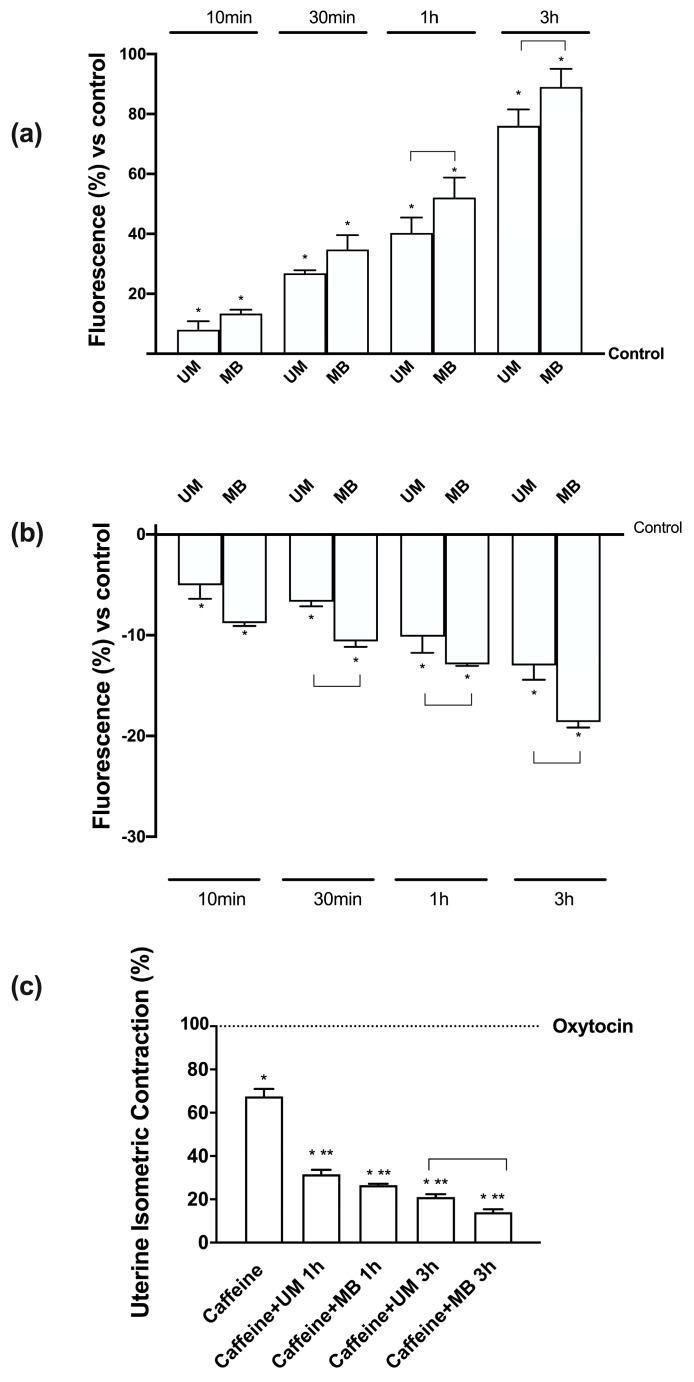
Measurements of Mg^2+^ and Ca^2+^ flux on PHN1-41 cells and mouse contractility assay. (**a**) Free intracellular magnesium concentration ([Mg2+]i) as measured before (basal [Mg^2+^]i) and after Mg-loading by UM and MB. After measurement of the basal [Mg^2+^]i in Na- and Mg-free medium, Caco-2 cells were Mg-loaded by Furaptra, and post-loading [Mg^2+^]i was calculated from the values measured over time. (**b**) Free calcium concentration ([Ca^2+^]) as measured before (basal) and after Mg-loading by UM and MB. After measurement of the basal [Ca^2+^] in Na-, Ca-, and Mg-free medium, Caco-2 cells were loaded with Fura2, and post-loading [Ca^2+^] was calculated from the values measured over time. Values are means ± SD (%) of six single experiments. * *p* < 0.05 vs. control; the bars indicate *p* < 0.05 between UM and MB at the same time point. (**c**) Analysis of contractility assay on mouse strips is reported. Data are expressed as means ± SD of four experiments normalized to the maximal contraction obtained using 10^−6^ M oxytocin. * *p* < 0.05 vs. oxytocin; ** *p* < 0.05 vs. caffeine; the bars indicate *p* < 0.05 between UM and MB at the same time point. The abbreviations are the same as reported in Figure 1.

**Figure 7 nutrients-12-00573-f007:**
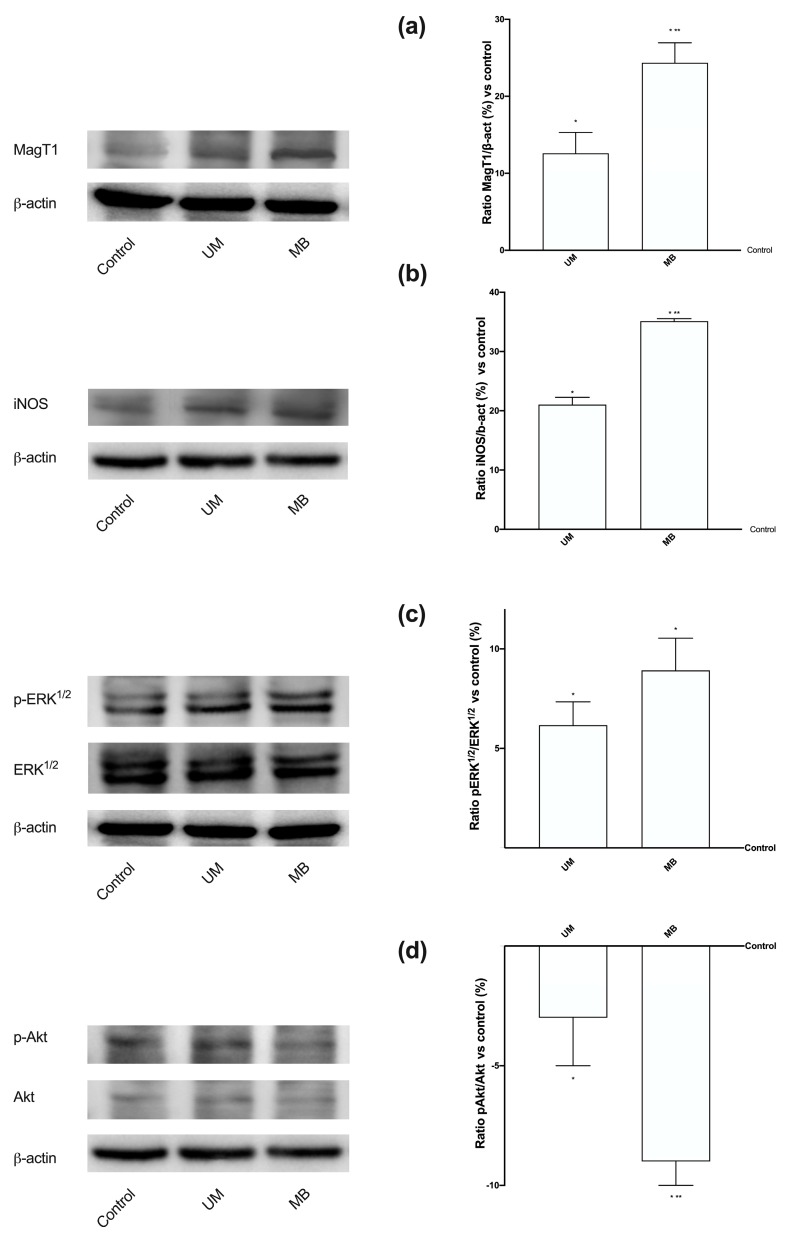
Western blot and densitometric analysis of some important kinases activated on PHM1-41 cells. Western blot and densitometric analysis of MagT1 (**a**), inducible nitric oxide synthase (iNOS) (**b**), ERK/MAPK (**c**), and PI3K/Akt (**d**), analyzed on whole PHN1-41 lysates at 3 h of stimulations. The images reported are an example of each protein from five independent experiments reproduced in triplicate. Data are expressed as means ± SD (%) of five independent experiments normalized on specific total protein if possible and verified on β-actin detection. * *p* < 0.05 vs. control; ** *p* < 0.05 vs. UM. The abbreviations are the same as reported in Figure 1.

**Figure 8 nutrients-12-00573-f008:**
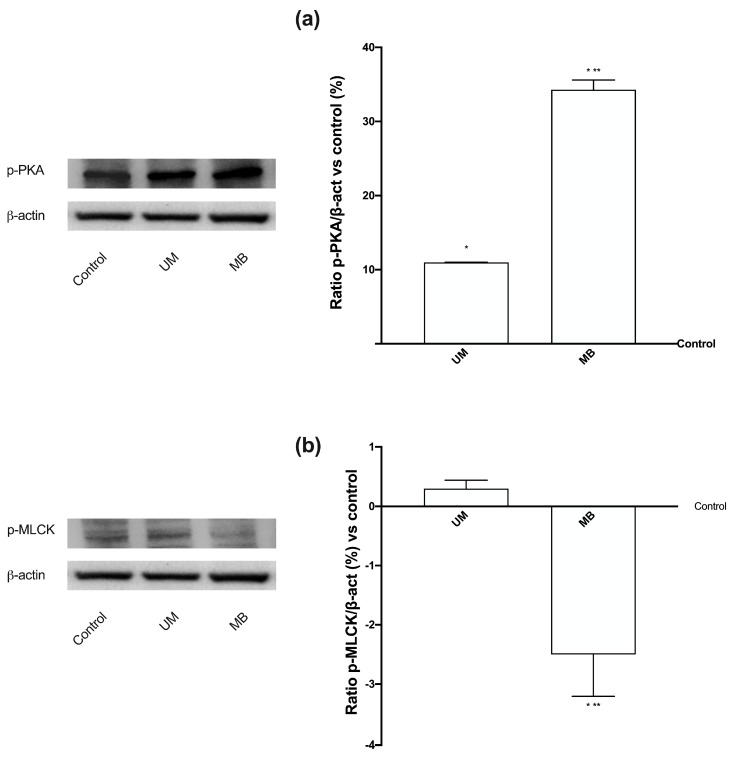
Western blot and densitometric analysis of PKA and phospho-myosin light chain (MLCK) on PHN1-41 cells. Western blot and densitometric analysis of PKA (**a**) and p-MLCK (**b**), analyzed on whole PHN1-41 lysates at 3 h of stimulations. The images reported are an example of each protein from five independent experiments reproduced in triplicate. Data are expressed as means ± SD (%) of five independent experiments normalized and verified on β-actin detection. * *p* < 0.05 vs. control; ** *p* < 0.05 vs. UM. The abbreviations are the same as reported in Figure 1.

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
