# Peer review of "Study of Magnesium Formulations on Intestinal Cells to Influence Myometrium Cell Relaxation"

_nutrients, 2020, doi:10.3390/nu12020573_

Round 1

Reviewer 1 Report

Two chelated Mg compounds are compared under in-vitro conditions without describing their chemical composition or discussing their mode of action. Despite 54 references (their presentation must be improved) two papers dealing with TrocaMag and UltraMag are not cited (Supakatisant et al: Matern Child Nutr 2015; 11:139, resp. Brilli E et al. Eur Rev Med Pharmacol Sci 2018; 22: 1843), the latter comparing UltraMag with the citrate and oxide in rats and volunteers. - On the other hand well known facts are described in length not being relevant in this context.

The experiments and results are of some interest. The authors should be encouraged to restrict to these data, i.e.,to shorten their manuscript, and to  discuss to what extent their results can be extrapolated to in-vivo conditions, resp. what data are needed for such purpose

Author Response

Two chelated Mg compounds are compared under in-vitro conditions without describing their chemical composition or discussing their mode of action.

Thank you for the comment. We apologize for the missing information. We add on “experimental protocol” in Method section some additional information about the chemical compositions of MB and UM). We have tested the biological effects but we didn’t investigate the chemical aspect since these Magnesium forms are commercially available along with specific chemical datasheet.

Despite 54 references (their presentation must be improved) two papers dealing with TrocaMag and UltraMag are not cited (Supakatisant et al: Matern Child Nutr 2015; 11:139, resp. Brilli E et al. Eur Rev Med Pharmacol Sci 2018; 22: 1843), the latter comparing UltraMag with the citrate and oxide in rats and volunteers. - On the other hand well known facts are described in length not being relevant in this context.

Thank you for the comment. We have taken into consideration the two refs you mentioned. The first describes that oral magnesium bisglycinate supplement can improve the frequency and intensity of pregnancy-induced leg cramps. In the second paper, where researchers focused only on the absorption and bioavailability capacities of various forms of Mg, Mg BIS showed a blood concentration comparable to that of Mg Sucrosomial and other Mg formulations.

Therefore, these data support the rationale of our study which aimed to study the cellular mechanisms, at the muscular level, of the various preparations containing Mg. We added in the manuscript the two references suggested by the reviewer.

The experiments and results are of some interest. The authors should be encouraged to restrict to these data, i.e.,to shorten their manuscript, and to  discuss to what extent their results can be extrapolated to in-vivo conditions, resp. what data are needed for such purpose

Thank you for the comment. A large number of tests were performed in this study and therefore a lot of data was collected. Therefore, it is not easy to reduce the results without penalizing the overall balance of the manuscript. In any case, we added some reasoning in discussion regarding the translationality of the results.

The Caco-2 cell model we use allows us to calculate permeability values that correlate well with human in vivo absorption data for several drugs and chemicals following the modern concept of translational medicine. As a result, the use of the Caco-2 model to predict human oral absorption of substances and ingredients has grown, as has its importance as a quality screening tool for intestinal absorption, especially in pharmacological research.

Thank you for the opportunity you gave us to revise the manuscript. All changes are reported in red in the manuscript.

Reviewer 2 Report

The authors have consistently outlined in the introduction of the manuscript the research subject and the recent developments and flaws still existing. However, I wonder whether this issue could be connected with preeclampsia as the MgSO4 is also a therapy for such process, that is, maybe the effects on the myometrium relaxation and the activity towards blood pressure could be correlated with a similar mechanism. The experiments are well-designed with the required information in case any colleague would reproduce the protocols. However, it should be necessary to indicate the passage number of the cells. Accordingly, the authors should clarify the selected concentration for MB and UM treatments (1 mM) and why did not perform a dose-dependent study (point 3.1 in results). Moreover, the authors applied the in vivo study with 9 experimental animals but there should be a pilot study or statistical analysis to determine the number of animals to obtain significant data. Another point to clarify is why the authors made the dose-response and time-dependent studies on cell viability in PHM1-41 cells with 1mM. Additionally, it is useful to the readers to have a AUC (area under the curve) data in order to estimate the total assimilation of Mg+2 in both the intracellular and transport studies.

Quality of the figures should be improved by increasing the size of the lettering in the axes

Author Response

The authors have consistently outlined in the introduction of the manuscript the research subject and the recent developments and flaws still existing. However, I wonder whether this issue could be connected with preeclampsia as the MgSO4 is also a therapy for such process, that is, maybe the effects on the myometrium relaxation and the activity towards blood pressure could be correlated with a similar mechanism.

Very recent data suggest that maternal Mg supplementation in pregnancy may have other perinatal benefits. A retrospective study of medical records reported that Mg supplementation during pregnancy was associated with a reduced risk of retarded fetal growth and preeclampsia. In addition, there is a lot of evidence that Mg supplementation during pregnancy may lead to prevent pregnancy complications and improves many health indicators and pregnancy outcomes [Zarean E, Tarjan A. Effect of Magnesium Supplement on Pregnancy Outcomes: A Randomized Control Trial. Adv Biomed Res. 2017 Aug 31;6:109. doi:10.4103/2277-9175.213879.]. In particular, in 10 trials, the composition of the magnesium supplements, was analyzed and no significant difference in perinatal mortality were reported. These studies on the different forms of magnesium have observed that the main differences concern the bioavailability (better with bysglicinate than MgSO4) and the cost (MgSO4 is cheaper than Mg BIS). Since in literature is well described the importance of Mg bioavailability during pregnancy, we hypothesize that the use of Mg BIS in the same conditions in which MgSO4 is used, may bring significant advantages.

 The experiments are well-designed with the required information in case any colleague would reproduce the protocols. However, it should be necessary to indicate the passage number of the cells.

Thank you for the comment. We apologise for the inconvenience and we added the lacking information about passage number of the cells in Method section (as reported in red).

Accordingly, the authors should clarify the selected concentration for MB and UM treatments (1 mM) and why did not perform a dose-dependent study (point 3.1 in results).

Thank you for the comment. We performed time-course and dose-dependent experiments on Caco-2 cells in order to identify the dose and time to use in all other experiments. In addition, a time course study was carried out also on PHM1-41. All These data about both cell types are reported in Table A1 and Table A2 present in Appendix B.

Moreover, the authors applied the in vivo study with 9 experimental animals but there should be a pilot study or statistical analysis to determine the number of animals to obtain significant data.

Thank you for the comment. The animals in this study were used to obtain and prepare the strips in which we can perform the contractility assay. The number of animals used was obtained starting from the data present in literature in which it is reported that from 1 animal we can obtain 4 strips [Yin Z, Sada AA, Reslan OM, Narula N, Khalil RA. Increased MMPs expression and decreased contraction in the rat myometrium during pregnancy and in response to

prolonged stretch and sex hormones. Am J Physiol Endocrinol Metab. 2012 Jul

1;303(1):E55-70. doi: 10.1152/ajpendo.00553.2011.]. We have calculated the number of sample’s stimulation and we used 9 animals in order to have 36 total strips. The use of 36 strips was assessed adequate by a preliminary statistical analysis (G*Power software). We added a sentence on the manuscript.

Another point to clarify is why the authors made the dose-response and time-dependent studies on cell viability in PHM1-41 cells with 1mM.

Thank you for the comment. We have decided to test the same concentration used on Caco-2 since we have also prepared a co-culture system in which we have added to Caco-2 1mM and we hypothesize that PHM1-41 cells received this concentration but pre-digested. In addition, in literature it is a concentration reported to have a beneficial effect (ref 11 on the manuscript)

Additionally, it is useful to the readers to have a AUC (area under the curve) data in order to estimate the total assimilation of Mg+2 in both the intracellular and transport studies.

Thank you for the comment. We have modified the graphic aspect of some figures (figs. 1 and 5) in order to highlight the difference between the AUC of the various Mg samples.

Quality of the figures should be improved by increasing the size of the lettering in the axes

Thank you for the comment. We changed all lettering in the axes of figures.

Thank you for the opportunity you gave us to revise the manuscript. All changes are reported in red in the manuscript

Round 2

Reviewer 1 Report

Insert a short comment on the two substances used; origin? dissoved in destilled water - or?

Author Response

Reviewer's comments 1:
Insert a short comment on the two substances used; origin? dissoved in destilled water - or?

Thank you for the comment. We apologize for the missing information. We add on “experimental protocol” in Method section the information required. Both magnesium forms were used directly in the medium without adding other agents.

A check of the English language was performed and typos were corrected.

Thank you for the opportunity you gave us to revise the manuscript. All changes are reported in red in the manuscript.